# Effect of Docosahexaenoic Acid (DHA) at the Enteric Level in a Synucleinopathy Mouse Model

**DOI:** 10.3390/nu13124218

**Published:** 2021-11-24

**Authors:** Jérôme Lamontagne-Proulx, Katherine Coulombe, Fadil Dahhani, Mélissa Côté, Cédric Guyaz, Cyntia Tremblay, Vincenzo Di Marzo, Nicolas Flamand, Frédéric Calon, Denis Soulet

**Affiliations:** 1Centre de Recherche du CHU de Québec, Québec, QC G1V 4G2, Canada; jerome.lamontagne-proulx.1@ulaval.ca (J.L.-P.); katherine.coulombe@vrr.ulaval.ca (K.C.); melissa.cote8.ciussscn@ssss.gouv.qc.ca (M.C.); Cyntia.Tremblay@crchudequebec.ulaval.ca (C.T.); frederic.calon@crchudequebec.ulaval.ca (F.C.); 2Faculté de Pharmacie, Université Laval, Québec, QC G1V 0A6, Canada; cedric.guyaz.1@ulaval.ca; 3Centre de Recherche de l’Institut Universitaire de Cardiologie et de Pneumologie de Québec (IUCPQ), Québec, QC G1V 4G5, Canada; fadil.dahhani@criucpq.ulaval.ca (F.D.); vincenzo.dimarzo@criucpq.ulaval.ca (V.D.); Nicolas.Flamand@criucpq.ulaval.ca (N.F.); 4Canada Excellence Research in the Microbiome-Endocannabinoidome Axis in Metabolic Health (CERC-MEND), Québec, QC G1V 4G5, Canada; 5Faculté de Médecine, Université Laval, Québec, QC G1V 0A6, Canada; 6Institut sur la Nutrition et les Aliments Fonctionnels (INAF) et Centre NUTRISS, Université Laval, Québec, QC G1V 0A6, Canada; 7Laboratoire International Associé OptiNutriBrain, (NutriNeuro France-INAF Canada), Québec, QC G1V 0A6, Canada

**Keywords:** Parkinson’s disease, DHA, synucleinopathy, enteric nervous system, glucagon-like peptide-1, Nrf2, Thy1-αSyn

## Abstract

The aggregation of alpha-synuclein protein (αSyn) is a hallmark of Parkinson’s disease (PD). Considerable evidence suggests that PD involves an early aggregation of αSyn in the enteric nervous system (ENS), spreading to the brain. While it has previously been reported that omega-3 polyunsaturated fatty acids (ω-3 PUFA) acts as neuroprotective agents in the brain in murine models of PD, their effect in the ENS remains undefined. Here, we studied the effect of dietary supplementation with docosahexaenoic acid (DHA, an ω-3 PUFA), on the ENS, with a particular focus on enteric dopaminergic (DAergic) neurons. Thy1-αSyn mice, which overexpress human αSyn, were fed ad libitum with a control diet, a low ω-3 PUFA diet or a diet supplemented with microencapsulated DHA and then compared with wild-type littermates. Our data indicate that Thy1-αSyn mice showed a lower density of enteric dopaminergic neurons compared with non-transgenic animals. This decrease was prevented by dietary DHA. Although we found that DHA reduced microgliosis in the striatum, we did not observe any evidence of peripheral inflammation. However, we showed that dietary intake of DHA promoted a build-up of ω-3 PUFA-derived endocannabinoid (eCB)-like mediators in plasma and an increase in glucagon-like peptide-1 (GLP-1) and the redox regulator, Nrf2 in the ENS. Taken together, our results suggest that DHA exerts neuroprotection of enteric DAergic neurons in the Thy1-αSyn mice, possibly through alterations in eCB-like mediators, GLP-1 and Nrf2.

## 1. Introduction

Parkinson’s disease (PD) is a common progressive neurological disorder that causes motor and non-motor dysfunctions [1]. Although the onset and intensity of symptoms are different for each patient, motor dysfunctions such as bradykinesia, resting tremor, and rigidity are typically observed [2]. It has been shown that pathological damage leading to a moderate or severe loss of neurons in the substantia nigra pars compacta (SNpc) underlies motor symptoms [3,4]. Indeed, one of the major pathological features of PD is defined by the loss of dopaminergic (DAergic) neurons in the SNpc [5,6]. Degeneration of DAergic projections from the SNpc towards the striatum causes a significant decrease in dopamine [7]. The connection between the ventrolateral SNpc and the dorsal region of the striatum, called the nigrostriatal pathway, is probably the most affected, resulting in 50 to 70% neuronal loss at diagnosis [5]. Neurodegeneration of the nigrostriatal pathway and accumulation of Lewy bodies in the SNpc are considered the gold standard in the diagnosis of PD [3]. Lewy bodies and Lewy neurites contain high amounts of aggregated hyperphosphorylated alpha-synuclein (αSyn) [8,9,10]. Research has shown that the locus containing the αSyn gene (SNCA) could be, in some cases, duplicated and even triplicated in some familial and sporadic cases of PD [11,12,13]. It is notable that the expression of mutations in this autosomal dominant gene seems to cause early development with rapid progression of the disease and prevalence for psychiatric disorders [14].

Accumulation of αSyn has been observed in the gastrointestinal tract (GIT) in PD patients, namely Lewy body pathology and positive αSyn immunohistochemistry in the myenteric and submucosal plexus of the enteric nervous system (ENS) [15,16,17]. Symptoms, identified as commonly occurring in PD, include constipation, nausea, decreased intestinal motility, and bloating. Constipation is probably the most important non-motor symptom that can precede motor dysfunctions by 15 years [18]. In 2003, Braak et al. hypothesized that the pathology related to αSyn could start in the ENS, possibly through toxins. The vagus nerve would allow the spread of the pathology from the ENS to the central nervous system [19,20]. During the last decade, series of experiments focusing on this hypothesis have provided interesting results. Indeed, human PD brain lysate containing αSyn injected into the rat ENS led to the accumulation and long-distance spread of the protein to the brain [21]. Furthermore, alteration of the gut has been demonstrated in most PD models. Lai et al., recently showed changes in microbiota composition, damage to the intestinal barrier, inflammation and alteration of enteric DAergic neurons in a toxin-induced mouse model of PD [22]. In addition, preformed fibrils injection models and transgenic models overexpressing αSyn demonstrated reduction in fecal water content, colonic transit time impairment and enteric accumulation of proteinase-K insoluble αSyn [23,24,25]. Further research into the involvement of the ENS in PD could lead to the discovery of new therapeutic treatments for the prodromal phase of the disease.

At present, pharmacological interventions can only attenuate motor and non-motor dysfunctions without stopping or slowing disease progression [26]. Therefore, it is critical to find natural, or synthetic compounds able to protect DAergic neurons. Several studies have shown the importance of diet on the development and evolution of PD [27,28], especially with diets rich in omega-3 polyunsaturated fatty acids (ω-3 PUFA), such as docosahexaenoic acid (DHA) [29]. DHA represents 10% of the total fatty acids and 40% of total PUFA in the brain [30,31], and is primarily known for its anti-inflammatory and anti-oxidant actions [32,33,34]. A diet rich in DHA was also shown to exert neuroprotective and neurorestorative effects in murine models of PD [29,35,36]. Diets enriched in DHA were demonstrated to prevent the decline of dopamine in the striatum as well as the loss of DAergic neurons in the SNpc in mouse models [37,38]. A recent publication also reported the involvement of ω-3 PUFAs in the reduction of PD motor and non-motor symptoms as well as αSyn overexpression using the functionality of the gut–brain axis [39].

In this study, we used Thy1-αSyn mice, which overexpress human αSyn (h-αSyn) under the Thy1 promoter. Of particular interest in this model is that the accumulation of h-αSyn observed in several regions of the brain [40]. In addition, studies reported various phenotypes, including motor deficits and a reduction in fecal pellet output without alteration of colon tyrosine hydroxylase-positive (TH^+^) nerve fibers in this transgenic model [25,41]. Therefore, this serves as a useful model in which to study the disease development with overexpression of the h-αSyn protein [40]. A previous study using this model showed that dietary DHA intake increased the longevity of Thy1-αSyn mice without improving motor deficits and with limited effects on DAergic neurons of the SNpc [42]. We hypothesized that DAergic neurons in the gut are more susceptible to synucleinopathy than in the brain and that a DHA-enriched diet would prevent the progression of PD by targeting DAergic neurons in the gut more efficiently than those in the SNpc. Thus, we collected gastrointestinal tissue and plasma from Thy1-αSyn mice that had been treated with a control, a low ω-3 PUFA or a DHA-enriched diet [42]. We specifically focused on the impact of fatty acids on the DAergic system in the ENS, the gut-immune response and the accumulation of h-αSyn.

## 2. Materials and Methods

### 2.1. Animals and Diets

Male Thy1-αSyn mice were bred in our animal facility from Thy1.2-αSyn mice (line 61). Thy1-αSyn and C57BL/6J littermates (wild-type) were housed in ventilated cages, on a 12 h/12 h dark/light cycle. Animals had free access to water and appropriate food. Dietary treatments were composed by a control diet (1.43 g/kg of ω-3 PUFA which includes 0 g/kg of DHA), a low ω-3 PUFA diet (0.31 g/kg of ω-3 PUFA which includes 0 g/kg of DHA) and a DHA-enriched diet (7.80 g/kg of ω-3 PUFA which includes 7.20 g/kg of DHA), detailed in Appendix A (taken from Coulombe et al., 2018) [42]. Purified diets formulas were designed by us, but produced by Research Diet Inc. (New Brunswick, USA). They are precisely determined to avoid batch-to-batch variations commonly found in laboratory chow. They are formulated to always contain the same amount of fat, protein, carbohydrates, fibers, vitamins, minerals, antioxidants, etc., without phytoestrogens, pesticides, or other common contaminants. In addition, we use microencapsulated DHA for its protection against oxidation and we did not use gamma irradiation or autoclaving on diets to avoid deterioration of the dietary treatment. All experiments were approved by the animal research committee of the Centre de recherche du CHU de Québec—Université Laval and performed according to the Canadian Guide for the Care and Use of Laboratory Animals. Mice were fed with a control, a low ω-3 PUFA, or a DHA-enriched diet for a ten-month period following weaning. DHA was obtained as a microencapsulated formulation to avoid oxidation (DSM Nutritional Product) and pellets were processed by Research Diets Inc. (New Brunswick, NJ, USA). We used only purified diet formulations that were standardized to ensure consistency and to eliminate batch-to-batch variation. Both diets were isocaloric and contained similar concentrations of macronutrients, vitamins and minerals. The dose of DHA was approximately 0.8 g/kg/daily per animal [42], which corresponds to 2–3 g/daily used in clinical trials [43,44]. All animals were sacrificed at 12 months of age by intracardiac perfusion with phosphate-buffered saline (PBS) containing protease and phosphatase inhibitors, performed after deep anaesthesia with a ketamine and xylazine mixture (respectively, 100 mg/kg and 10 mg/kg). This project involved the use of murine tissues collected during a previous study [42]. The blood was collected and centrifuge at 2500× *g* for 15 min, then plasma was harvested and stored at −80 °C. The small intestine was collected and post-fixed in 4% paraformaldehyde (PFA) for 48 h, then stored in a solution composed with 20% sucrose and 0.05% sodium azide in 0.1 M PBS pH 7.4 at 4 °C. Five to six sections (1–10 mm^2^/section) of small intestine sampled in the distal ileum were microdissected to reveal the myenteric plexus (or Auerbach’s plexus) hidden between muscle layers as previously described [45].

### 2.2. Immunofluorescence

Immunoreactivity quantifications were performed by immunofluorescence. Myenteric plexus and brain sections were incubated for 1 h at 100 °C in sodium citrate for antigen retrieval before a 30-min blocking treatment with a solution of 0.4% Triton X-100 and 5% donkey serum (Sigma-Aldrich) in PBS 1×. Free-floating tissues were stained overnight or 5 days (GFAP only) with different primary antibodies, followed by a 2-h incubation with secondary antibodies in blocking solution. Counterstaining with 0.022% DAPI (Invitrogen Corporation, Waltham, MA, USA) was performed before mounting sections. Refer to Appendix A for antibody listings.

MHCII- and Iba1-labelled sections were imaged using a Zeiss AxioScan.Z1 scanner and Zen 2.3 acquisition software (Carl Zeiss Canada, Toronto, ON, Canada). All other samples were imaged as previously described [46] using an Olympus IX81-ZDC FV1000 confocal microscope (Olympus Canada, Richmond Hill, ON, Canada). Three microdissected myenteric plexus sections were imaged per animal with three images per sections. For h-αSyn, GLP-1 and GPR120 analyzes, ChAT was used as a reference for sections of the tissue to be imaged. Mean pixel intensity was determined on images for immunoreactivity quantification, and for density count, labeled cells were calculated as the number of positive cells per area (mm^2^), using NIH Fiji software. Three-dimensional reconstruction from Z-stack (Figure 1B) were generated with Bitplane Imaris software v7.6.5 (Bitplane, Zurich, Switzerland). Mean values were calculated with the microdissected sections for each animal. All images were captured, and data analyses undertaken, blindly.

### 2.3. Cytokines Immunoassay

Cytokines levels in plasma of the Thy1-αSyn mice and their wild-type littermates were quantified using a Bio-Plex Pro Mouse Cytokine 8-Plex Assay kit (#M60000007A, BioRad, Hercules, CA, USA) according to manufacturer’s instructions. 25 μL of each diluted (1:2) plasma sample was used for immunoassay analysis. Based on standard curves, IL-1β, IL-5, IL-10, IFN-γ and TNF-α values were determined and adjusted according to the dilution. While IL-4 was only detectable in too few samples, IL-2 and GM-CSF were not detectable at all. 

### 2.4. Endocannabinoidome Mediators

Lipid mediators were quantified using standard isotope dilution and the appropriated deuterated internal standards (Appendix A). They were quantified in 40 µL of plasma as described by Turcotte et al. [47]. In brief, Tris (50 mM, pH 7) was added to plasma samples to a final volume of 500 µL. Toluene (2 mL) containing the ISTD was added to the samples, vortexed for 1 min, centrifuged at 4000× *g* for 5 min without brakes. Samples were next placed in an ethanol-dry-ice bath (−80 °C) to freeze the aqueous phase (bottom). The organic (upper) phase was then collected and evaporated to dryness under a stream of nitrogen. Samples were reconstituted in 25 µL of HPLC solvent A (H_2_O with 0.05% acetic acid and 1 mM NH_4_^+^) and 25 µL of solvent B (MeCN/H_2_O, 95/5, *v*/*v*, with 0.05% acetic acid and 1 mM NH_4_^+^). 25 µL of the resulting mixture was injected onto an RP-HPLC column (Kinetex C8, 150 mm × 2.1 mm, 2.6 µm, Phenomenex, Torrance, CA, USA). Quantification was performed on a Shimadzu 8050 triple quadrupole mass spectrometer using the same LC program as described previously [48]. The method can differentiate monoacylglycerol isomers at positions 1 and 2 but signals from both isomers of unsaturated fatty acids were summed—and identified as 2-monoacylglycerols-prior to analysis in order to account for their rapid interconversion. However, similar results were obtained when analyzing either 1(3)- and 2-isomers combined or using only 2-monoacylglycerol signals. The following fatty acids and metabolites were quantitated: arachidonic acid (AA), anandamide (AEA), 1/2-arachidonoyl-glycerol (2-AG), docosahexaenoic acid (DHA), docosahexaenoyl-ethanolamide (DHEA), 1/2-docosahexaenoyl-glycerol (2-DHG), docosapentaenoic acid (n-3) (DPA), docosapentaenoyl-ethanolamide (n-3) (DPEA), docosapentaenoyl-glycerol (1/2-DPG), eicosapentaenoic acid (EPA), eicosapentaenoyl-glycerol (1/2-EPG), linolenic acid (LA), linoleoyl-ethanolamide (LEA), linoleoyl-glycerol (1/2-LG), oleoyl-ethanolamide (OEA), oleoyl glycerol (1/2-OG), palmitoyl-ethanolamide (PEA), stearidonic acid (SDA), stearoyl-ethanolamide (SDEA), stearidonyl-glycerol (2-SDG).

### 2.5. Statistical Analyses

Data were evaluated with the Shapiro–Wilk test to determine normality and the Brown-Forsythe test to evaluate equal variance. Since data were normally distributed and had equal variance, comparisons between groups were calculated by one-way analysis of variance (ANOVA) followed by Tukey’s post-hoc tests. Results for each animal is represented (white dots), as well as the mean ±SEM. Results are considered statistically significant if *p* < 0.05. All statistical analyses were performed with Prism 8 (GraphPad Software Inc., La Jolla, CA, USA) software.

## 3. Results

### 3.1. Polyunsaturated Fatty Acid (PUFA) Diets Modulate Plasma Fatty Acid Composition

Fatty acid composition analysis in plasma revealed the efficacy of the DHA diet to increase circulating levels of DHA, eicosapentaenoic acid (EPA), docosapentaenoic acid (ω-3 DPA) and stearidonic acid (SDA). Besides, DHA intake decreased both arachidonic acid (AA) and linolenic acid (LA), both ω-6 PUFA (Table 1). The opposite can be observed in animals fed the low ω-3 PUFA diet in which DHA, EPA and SDA plasma levels are lower (Table 1). Interestingly, Thy1-αSyn mice showed higher levels of LA and SDA compared to non-transgenic littermates (Table 1).

### 3.2. Neuroprotection of Enteric DAergic Neurons by Docosahexaenoic Acid (DHA) 

Two types of myenteric plexus neurons were quantified to assess differences between wild-type mice and the Thy1-αSyn model and to determine the impact of DHA intake. Quantification of TH^+^ DAergic neurons revealed a lower density in Thy1-αSyn mice compared with non-transgenic littermates (*p* = 0.0203; Figure 1A). Of interest is that the Thy1-αSyn mice fed with the DHA-enriched diet shows a higher density of DAergic neurons compared with the control diet mice (*p* = 0.0037; Figure 1A). This high density was not observed in animals that received the low ω-3 PUFA diet (*p* = 0.0490 compared with non-transgenic mice; *p* = 0.0116 compared with Thy1-αSyn mice maintained on the DHA-enriched diet; Figure 1A). Furthermore, the DHA-enriched diet increased the number of TH^+^ varicosities in Thy1-αSyn mice (*p* = 0.0308; Figure 1B), although there were no differences between the model and non-transgenic mice. Thereafter, quantifications were conducted for cholinergic (ChAT+) neurons by immunoreactivity and results showed no significant difference between Thy1-αSyn and wild-type animals. Moreover, there was no effect of the DHA-enriched diet on CHAT+ neurons (Figure 1C). Overall, the comparison between mice genotype and diet revealed a lower level of TH^+^ DAergic neurons in the Thy1-αSyn model compared with the non-transgenic mice. Furthermore, these data show that the DHA-enriched diet increased the TH^+^ neuron density in the Thy1-αSyn model, to a level similar to that observed in the wild-type mice.

### 3.3. Validation of h-αSyn Expression in the Central and Enteric Nervous System

Brain sections were labeled using a rabbit antibody against wild-type h-αSyn (MJFR1) and a sheep antibody against TH^+^ DAergic neurons to validate the presence of αSyn in selected sections, with particular emphasis on the striatum and the SNpc. Fluorescence imaging confirmed the distribution of h-αSyn in the brain of the Thy1-αSyn mice and the absence of the protein in the non-transgenic mice (Appendix A). Afterwards, h-αSyn was labeled and quantified to determine a possible impact of DHA on the presence of the protein in the mouse myenteric plexus. Immunofluorescence using MJFR1 and the goat antibody against ChAT in the myenteric plexus of Thy1-αSyn mice revealed a high level of h-αSyn in this sublayer of the intestine (Figure 2A). h-αSyn was quantified on ChAT-positive images in order to avoid selection bias. As expected, no h-αSyn immunofluorescence was detected in wild-type mice (Figure 2B). Furthermore, no significant effect was observed for any dietary treatment in Thy1-αSyn mice with regard to mean gray intensity of h-αSyn, suggesting that DHA does not regulate h-αSyn accumulation (control diet vs. DHA-enriched diet, *p* = 0.9717) (Figure 2B). αSyn fluorescence did not appear to colocalize with ChAT+ neurons, as the protein was observed in close proximity to the varicosities (Figure 2A). However, photomicrographs of myenteric neurons labeled with αSyn, and TH antibodies show a colocalization between the protein and TH^+^ varicosities, as well as αSyn^+^/TH^−^ varicosity-like structure (Figure 2C). To conclude, these results appear to confirm the presence of h-αSyn in the ENS of Thy1-αSyn mice with no apparent diet-induced modulation.

### 3.4. Detection of G-Protein Coupled Receptor 120 (GPR120) in the Myenteric Plexus

Published evidence show that DHA could mediate its neuroprotective effect by binding to G-protein coupled receptor 120 (GPR120) [49,50]. To confirm the expression of the receptor in the myenteric plexus of both animal strains, we performed an immunofluorescence study with a rabbit antibody against GPR120 (Figure 3A). Immunoreactivity analysis of GPR120 showed no significant difference between non-transgenic mice and Thy1-αSyn mice, nor among dietary groups (Figure 3B). Immunofluorescence revealed the presence of GPR120 in the myenteric ganglia in both mouse model. Furthermore, the receptor colocalized with TH^+^ cell bodies in every group (Figure 3C, only Thy1-αSyn with control diet is shown). These results indicate that DHA via GPR120 could contribute to different functions in the ENS and directly in enteric DAergic neurons.

### 3.5. Thy1-αSyn Mice Show Inflammation at the Central Level but Not in Periphery 

A previous study shows that Thy1-αSyn mice display inflammation indexed with microglia density (Iba1+) at 5 months in the central nervous system [40]. To investigate the immunomodulatory role of DHA, we performed immunofluorescence analysis of astrocytes (GFAP+) and microglia (Iba1+) in the striatum of non-transgenic littermates and Thy1-αSyn mice. No differences in astrocyte activation were observed (Appendix A). However, results show increased microglial activation in the transgenic model compared to wild-type mice. Moreover, Thy1-αSyn animals fed with the low ω-3 PUFA or the DHA-enriched diets had lower level of microglia compared to control diet, suggesting anti-inflammatory effect (Appendix A). To explore inflammation at the myenteric level, we analyzed the density of total macrophages (Iba1+) (Figure 4A) and pro-inflammatory macrophages (MHCII+) (Figure 4C). No difference in density was observed between mouse strains or as a function of diet for both cells type (Figure 4B,D). We further investigated plasma cytokines IL-1β, IL-4 (not detectable), IL-5, IL-10, IFN-γ and TNF-α. No difference was observed between mouse strains and diets for all cytokines (Figure 4E–I). Thus, our results suggest that DHA decrease pro-inflammatory microglia density in brain. Since our results show no inflammation in periphery, DHA has neither an effect on enteric macrophages nor on plasma cytokines in 12-month-old mice.

### 3.6. DHA Promotes Glucagon-Like Peptide 1 (GLP-1) Production

Previous observations suggest that long chain fatty acids, like DHA, stimulate glucagon-like peptide 1 (GLP-1) secretion through GPR120 [51]. To confirm the diet-dependent secretion of GLP-1 in our model, we performed an immunofluorescence study with rabbit antibody against GLP-1 (Figure 5A). Immunoreactivity analysis revealed a significantly lower level of GLP-1 in Thy1-αSyn mice compared with non-transgenic littermates, both on the control diet (*p* = 0.0346; Figure 5B). As expected, the DHA-enriched diet increased the concentration of GLP-1 in the transgenic mice compared with mice maintained on the control diet (*p* = 0.0069; Figure 5B). A non-significant increase in GLP-1 was observed with the low ω-3 PUFA diet, suggesting a reduced effect by other PUFAs on GLP-1 secretion compared with DHA (Figure 5B). These results support that DHA significantly increases GLP-1 release compared with the control diet.

### 3.7. Decrease in Nrf2 in Thy1-αSyn Mice Is Prevented by the DHA-Enriched Diet

Oxidative stress is an important pathological feature detected in PD patients and altered gene expression regulating antioxidative responses could lead to cell death [52]. To assess the ability of enteric neurons to defend themselves against oxidative stress, we used immunofluorescence to determine the quantity of transcription factor Nrf2 (nuclear (erythroid-2) related factor-2) inside the neuron cytoplasm (Figure 6A). Cytoplasmic-Nrf2 was quantified inside the neuronal specific HuC/HuD (ELAV-like protein 3 and 4)-positive cell body volume. Results showed that Thy1-αSyn mice maintained on control and low ω-3 PUFA diets had a lower level of cytoplasmic-Nrf2 protein in enteric neuron bodies compared with wild-type animals (*p* = 0.0419 for control diet and *p* = 0.0378 for low ω-3 PUFA diet) (Figure 6B). Conversely, transgenic animals maintained on the DHA-enriched diet presented a higher level of the transcription factor compared with Thy1-αSyn mice fed with the control and low ω-3 PUFA diets in both cytoplasm (*p* = 0.0286 for control diet and *p* = 0.0250 for low ω-3 PUFA diet) (Figure 6B). These findings suggest that DHA can increase Nrf2 levels and thereby, reduce oxidative stress.

### 3.8. Modulation of Plasma Endocannabinoid-like Mediators Following PUFA Diets

An increasing amount of evidence suggest that bioactive metabolites of DHA have multiple beneficial effects including modulation of immune response and neuroprotection [53,54]. Among these, endocannabinoid-like (eCB-like) mediators display anti-inflammatory, neurite growth and synaptogenesis properties [55]. Analysis of N-acylethanolamine (NAE) and monoacylglycerols (MAG) profiles in plasma revealed that a high DHA diet increases ω-3 PUFA-derived eCB-like metabolites and decreases ω-6 PUFA-derived metabolites (Table 2). Furthermore, an increase in linoleoyl glycerol (LG) and oleoyl glycerol (OG) levels has been observed in Thy1-αSyn mice compared to non-transgenic animals (Table 2). These results support the idea that eCB-like mediators in plasma can be modulated by the PUFA composition of diets.

## 4. Discussion

The aim of this study was to evaluate the effects of a dietary DHA supplementation on DAergic neurons in the ENS. We show that the number of TH^+^ enteric DAergic neurons in 12-month-old Thy1-αSyn mice was reduced compared with wild-type littermates, and that this loss of TH^+^ neurons was prevented by a DHA-enriched diet. Our results are consistent with the following conclusions: Accumulation of h-αSyn has a greater deleterious impact on DAergic neurons in the ENS than in the brain in Thy1-αSyn mice [42] and DHA exerted a strong neuroprotective effect on DAergic cells in the ENS. Potential mechanisms for the neuroprotection include DHA-induced secretion of glucagon-like peptide-1 and Nrf2 as well as DHA-dependent modulation of beneficial eCB-like mediators.

The study conducted by Coulombe et al. included 30 wild-type and more than 45 Thy1-αSyn mice and showed a 13% loss of TH^+^ cell count in the SNpc of Thy1-αSyn mice compared with wild-type littermates [42]. Despite the widespread expression of h-αSyn and the loss of DAergic neurons in the brain, Thy1-αSyn mice showed similar levels of striatal DA compared with non-transgenic mice. Measures of DHA in the prefrontal cortex revealed higher level of this fatty acid in mice that received the DHA-enriched diet, confirming the efficacy of the dietary treatment [37,42]. Of interest is the observation that the DHA-enriched diet appears to partially protect the nigral neurons in addition to increasing dopamine release [42]. Otherwise, DHA had no effect on striatal dopamine, dopamine transporter (DAT) and TH expression in wild-type mice [37]. As expected, our analysis of plasma fatty acid composition revealed also higher level of DHA in animals fed with the DHA-enriched diet compared to control diet. Moreover, the results of our study suggest that the loss of TH^+^ cell bodies in the ileum is more substantial than the loss in the SNpc, with a decrease of 67% in the ENS of Thy1-αSyn mice compared with wild-type mice. Enteric TH^+^ neurons in Thy1-αSyn mice appear to be protected by the DHA-enriched diet, showing similar levels to wild-type littermates. This result is specific to the DHA-enriched diet since the low ω-3 PUFA diet had no effect. Furthermore, analysis of TH^+^ varicosities density shows a specific DHA synaptogenic effect. To evaluate the specificity of the TH^+^ neuron protection, cholinergic neurons were also quantified as they represent the highest population of neurons in the ENS [56,57]. No differences between animals or among dietary intakes were observed for this neuron type.

A previous study reported the presence of h-αSyn in the ENS of the Thy1-αSyn model, showing a 3-fold higher density of h-αSyn-positive neuronal fibers compared to wild-type mice [25]. We confirmed the presence of the the protein in the brain and in the ENS at the myenteric plexus level in Thy1-αSyn mice. H-αSyn seems to be present in varicosities, and more interestingly, in TH^+^ varicosities. However, quantification of h-αSyn expression revealed no difference among diets. These results suggests that DHA may not protect dopaminergic neurons by interacting with h-αSyn in this model. However, this fatty acid could have many other beneficial functions including anti-inflammatory and anti-oxidative properties [32,33,37,58,59].

To the best of our knowledge, our results are the first to show the presence of G-protein coupled receptor 120 (GPR120) in neurons of the myenteric plexus. Although we did not observe diet-dependent changes in immunostaining, a particularly interesting observation was the presence of GPR120 in the soma of enteric neurons, including TH^+^ dopaminergic neurons. Previous reports have shown its interaction with ω-3 PUFAs and it presence mainly in intestinal endocrine cells [49,50,60]. GPR120 is reported to play important roles in regulating metabolism, inducing anti-inflammatory effects and promoting intestinal peptide secretion [60,61,62]. In addition, a previous study reported the presence of the receptor in neurons of the hypothalamus to mediate anti-inflammatory actions [63]. Our finding supports a direct interaction between DHA and these enteric neurons, opening the possibility that the neuroprotective effect of DHA is mediated via GPR120 receptors.

DHA anti-inflammatory action have been shown to involve GPR120 activation and subsequent modulation of macrophage response and immunophenotype [49,64,65]. Previous studies of the Thy1-αSyn mice model show increased number of activated microglia detected in the striatum of 1-month-old mice and later in the SNpc of 5-month-old mice, persisting up to 14 months-old in both regions [40,66]. Supporting these results, we reveal here an increase of microglia activation in the striatum of 12 months-old Thy1-αSyn mice. Furthermore, recent studies reveals that DHA-enriched diet and LA-enriched diet can reduce microglia activation [67,68]. In accordance with these previous studies, transgenic animals on the DHA-enriched diet or on the low ω-3 PUFA diet (LA-enriched diet) showed a decrease in microgliosis. Thus, we investigated whether inflammatory mechanisms could be also involved in the periphery, and whether PUFA could have immunomodulatory properties in the gut and the blood.

Considerable evidence supports the concept that peripheral inflammation plays a critical role in the progression of PD [45,69,70,71]. Intestinal macrophages represent a heterogeneous population of immune cells that may play a major role in host defense, tissue regulation, and modulation of the inflammatory response [72,73]. Furthermore, increased levels of inflammatory markers are critical for immune activation and have been reported many times in PD patients [71,74]. To investigate the development of peripheral inflammation in the Thy1-αSyn model and the anti-inflammatory potential of DHA, we first quantified the density of total Iba1+ macrophages and the density of pro-inflammatory MHCII+ macrophages in the ileum. Our results show that Thy1-αSyn mice present no increase in macrophage activation or recruitment in the myenteric plexus. Next, we measured anti-inflammatory (IL-10) and pro-inflammatory (IL-1β, IL-5, IFN-γ and TNF-α) cytokines in plasma. We did not notice any differences for any cytokine between the transgenic model and wild-type mice. We cannot rule out that the lack of inflammation is specific to this mouse model at this time point. For example, in the 1-methyl-4-phenyl-1,2,3,6-tetrahydropyridine (MPTP) model, enteric macrophages and plasma cytokines may play a major role in the loss of enteric TH^+^ cell bodies [45,70,75]. Our results suggest that DHA could protect enteric neurons by mechanisms that are unrelated to inflammation in our model.

With regard to another important function of GPR120, this receptor has been shown to colocalize with GLP-1 in murine intestinal epithelial cells and human colonic intraepithelial cells [60,76]. Some studies have shown that stimulation of GPR120 by ω-3 PUFA intake, particularly DHA, could mediate GLP-1 secretion up to a two-fold level in plasma [51,60,77]. GLP-1 is an incretin hormone released from intestinal endocrine L cells that can be specifically stimulated by fibers, peptides, amino-acids and unsaturated fatty acids [78]. One of its many functions is to increase glucose metabolism after food intake, regulating glucose-dependent insulin secretion by pancreatic beta cells [79,80]. This intestinal peptide can also exert inhibitory effects on GIT motility, directly influencing the GIT wall and improve myenteric neuron survival [81,82]. Furthermore, GLP-1 and its analogues have been shown to modulate dopamine neurotransmission in different regions of the central nervous system [83,84]. Of interest, a GLP-1 receptor (GLP-1R) agonist has been reported as a potential treatment for PD [85]. GLP-1R agonism afforded complete protection against dopaminergic neurodegeneration and microglia activation in rodent models of PD, and GLP-1R agonists are now undergoing clinical trials [86,87,88]. A reduction in TH^+^ neurons and a decrease of brain dopamine content induced by MPTP were fully prevented by treatment with a GLP-1R agonist [89]. Activation of this receptor increased the number of viable dopaminergic neurons and induced TH gene transcription in the area postrema [90]. Our analysis of this incretin revealed that the DHA-enriched diet increased GLP-1 levels in the murine myenteric plexus. This study is the first to suggest that DHA could exert neuroprotective effects on enteric DAergic neurons. This effect could be mediated, at least partially, by GLP-1 (schematic representation, Figure 7).

Oxidative damage caused by free radicals, an important contributor to PD development [91], has already been observed in Thy1-αSyn mice. This model shows an impairment in mitochondrial function accompanied by an increase in oxidative stress in the ventral midbrain, suggesting an impairment in antioxidant defense that could participate in dopaminergic neuron degeneration [92]. It is interesting that GLP-1 has been shown to exert an antioxidative action, reducing oxidative stress markers by phosphorylation of Nrf2 [93,94]. The transcription factor Nrf2 and its binding site ARE (antioxidant response element) are the master regulators of redox homeostasis by modulating the expression of over 250 genes including superoxide dismutase, catalase and glutathione peroxidase [95]. In addition to regulation by GLP-1R, evidence suggests that ω-3 PUFAs, such as DHA, could activate Nrf2-ARE signaling directly [96,97,98]. Moreover, recent studies propose that the activation of GPR120 could increase Nrf2 levels [99,100]. Furthermore, greater nigral dopaminergic neuron loss and more prominent dystrophic dendrites are found in Nrf2 knockout animals following αSyn injection in the ventral midbrain [101]. To explore the modulation of Nrf2 in the ileum of Thy1-αSyn mice, the redox transcription factor was quantified in the cytoplasm of enteric neurons. Our observations revealed lower levels of cytoplasmic Nrf2 in Thy1-αSyn mice maintained on the control and low ω-3 PUFA diets compared with wild-type animals. However, the DHA-enriched diet increased the cytosolic Nrf2 levels compared with the other diets. These data show that DHA can significantly increase Nrf2 levels, consistent with a protective effect against oxidative stress (Figure 7). Of interest, specific Nrf2-dependent genes appear to be down regulated in PD patients despite an upregulation of Nrf2 in the SNpc [102]. Our results, combined with previous studies, suggest that the transcription factor Nrf2 could be an emerging target to protect DAergic neurons from oxidative damage in both the brain and ENS [103,104]. 

A large body of evidence suggests that various families of downstream metabolites contribute to the neuroactivity of dietary ω-3 PUFAs [53]. These metabolites include eCB-like mediators. The main examples ω-3 PUFA-containing are NAEs, such as docosahexaenoyl-ethanolamine (DHEA, also known as synaptamide) [105]. It is widely accepted, and shown in our study, that DHEA blood levels are increased following dietary intake of DHA in animals and humans [106,107,108]. The association between the eCB system and PD has been explored in the last two decades showing modulation of the action of some antiparkinsonian drugs [109,110]. However, the role in PD of eCB-like mediators such as DHEA is unknown. There is evidence that DHEA displays anti-inflammatory properties and ameliorates synaptic plasticity [55,111]. Thus, DHEA could participate in the decrease of microglia activation and the increase of enteric TH^+^ varicosities observed following the DHA-enriched diet. Additionally, higher levels of ω-6 PUFA-containing eCBs, such as anandamide (AEA) and arachidonoyl glycerol (2-AG), in either brain or cerebrospinal fluid are associated with PD symptoms in experimental models and humans [110,112], and antagonism of eCB action at CB1 receptors was found to ameliorate such symptoms [112,113], whereas activation of eCB CB2 receptors (at which anandamide has much lower affinity than 2-AG) was suggested to produce instead anti-inflammatory actions in PD models [114]. Therefore, it is also possible to hypothesize that reduction by dietary DHA of anandamide, but not 2-AG, levels in Thy1-αSyn mice is partly responsible for disease amelioration in this model.

Finally, we cannot exclude that DHA could also promote alternative mechanisms for neuronal protection (e.g., increased membrane fluidity, an increase in trophic factors, apoptosis inhibition, mitochondrial protection, anti-inflammatory pathways) [29]. Therefore, further studies are required for a more comprehensive investigation of the underlying mechanisms in order to provide evidence for the combinatorial use of several therapeutic agents.

## 5. Conclusions

In summary, this study provides evidence that dietary intake of DHA can protect enteric DAergic neurons from synucleinopathy-induced DAergic neurodegeneration. The comparison with a previous study performed in the same animals [42] revealed that the preventive effect of DHA was more pronounced in the ENS than in the brain. We are the first to propose that neuroprotection of DAergic neurons with DHA could be mediated by increases in ω-3 PUFA-containing eCB-like mediators, enhanced GLP-1 secretion as well as higher in Nrf2 levels in the ENS. Given the results observed suggesting a reduction in enteric neuronal damage, we suggest that DHA-mediated pathway could be potential therapeutic avenues for PD patients. Further studies are needed to understand the mechanisms of DHA action in the brain and the ENS.

## Figures and Tables

**Figure 1 nutrients-13-04218-f001:**
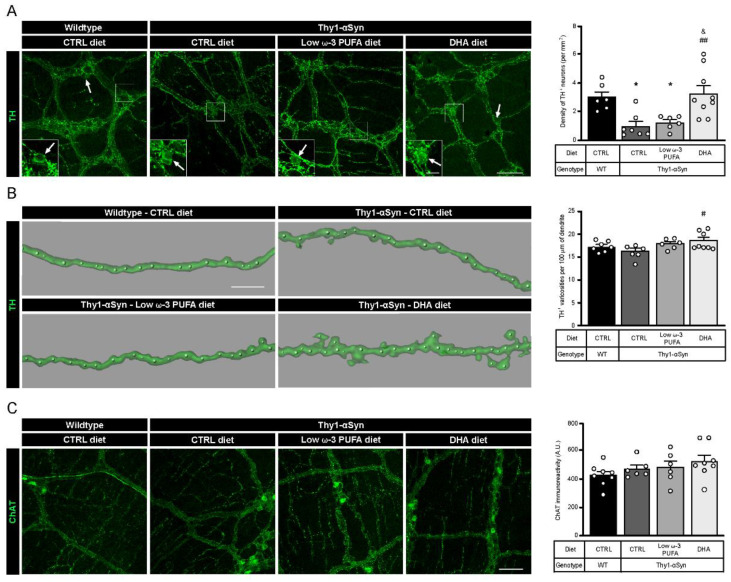
A DHA-enriched diet restores dopaminergic neuron integrity. (**A**) Photomicrographs of enteric TH^+^ neurons (green) and their cell body, as indicated by the white arrow. They show a lower number of dopaminergic cell bodies in the myenteric plexus in the Thy1-αSyn model compared with the wild-type littermates. However, a DHA-enriched diet protected against the decrease in dopaminergic neuron level (one-way analysis of variance (ANOVA): *p* = 0.0009). Scale bar = 60 µm; Close-up scale bar = 15 µm. (**B**) Three dimensional images of TH^+^ dendrites (green) and varicosities (white dots) show an increase of varicosities per 100 μm of dendrites in transgenic mice fed with DHA-enriched diet compared to control diet. No difference was observed between the Thy1-αSyn mice and the wild-type littermates (one-way ANOVA: *p* = 0.0396). Scale bar = 10 μm. (**C**) Photomicrographs and immunoreactive quantification of ChAT (green) shows no difference between models or among dietary groups (one-way ANOVA: *p* = 0.2987). Scale bar = 60 μm. Tukey’s post-hoc tests: * *p* < 0.05 compared with wild-type mice; ^#^
*p* < 0.05, ^##^
*p* < 0.01 compared with Thy1-αSyn on control diet; ^&^
*p* < 0.05 compared with Thy1-αSyn on low ω-3 PUFA diet. Abbreviations: A.U., arbitrary unit; ChAT, choline acetyltransferase; CTRL, control; DHA, docosahexaenoic acid; TH, tyrosine hydroxylase; WT, wild-type.

**Figure 2 nutrients-13-04218-f002:**
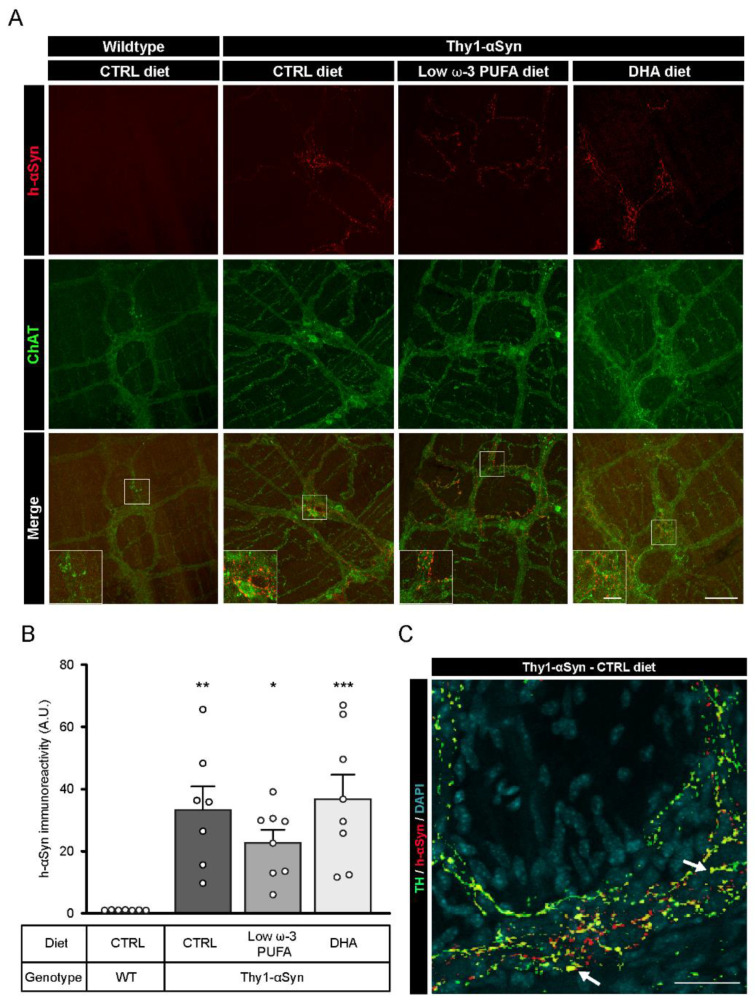
Presence of αSyn in the myenteric plexus of Thy1-αSyn mice. (**A**) Immunofluorescence of ChAT^+^ neurons (green) and αSyn (red) confirms the presence of human αSyn in the Thy1-αSyn model ENS and absence of the protein in the wild-type littermate. Scale bar = 60 µm; Close-up scale bar = 15 µm. (**B**) Quantification of wild-type human αSyn by immunoreactivity in the myenteric plexus reveals a higher level of the protein in Thy1-αSyn mice compared with a wild-type littermate, but no difference among diets (one-way ANOVA: *p* = 0.0005). Tukey’s post-hoc tests: * *p* < 0.05, ** *p* < 0.01 and *** *p* < 0.001 compared with wild-type mice. (**C**) Photomicrographs of the myenteric plexus showing TH^+^ neurons (green) and αSyn (red) reveal the presence of αSyn in TH^+^ varicosities (yellow), as indicated by the white arrow. Scale bar = 30 µm. Abbreviations: h-αSyn, human alpha-synuclein; A.U., arbitrary unit; ChAT, choline acetyltransferase; CTRL, control; DHA, docosahexaenoic acid; ENS, enteric nervous system; WT, wild-type. (white dots) show an increase of varicosities per 100 μm of dendrites in transgenic mice fed with DHA-enriched diet compared to control diet.

**Figure 3 nutrients-13-04218-f003:**
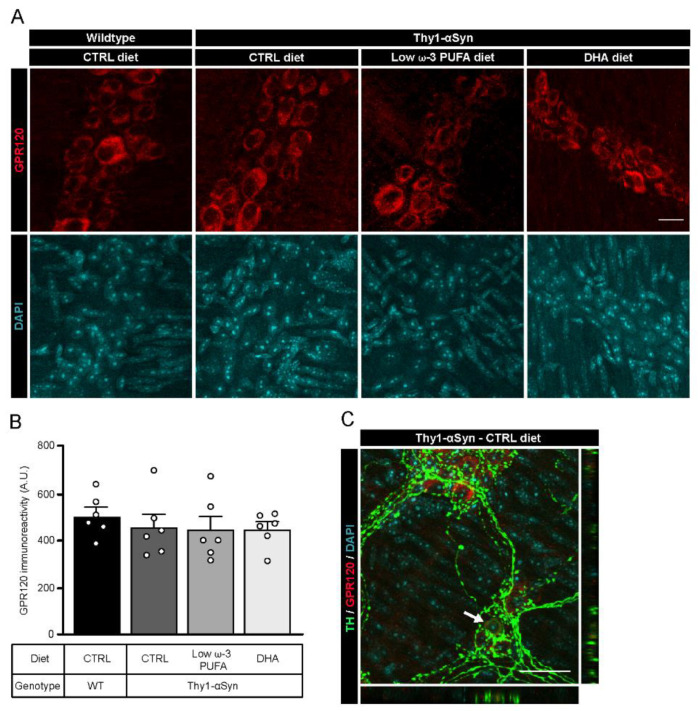
Localization of the DHA receptor GPR120 in the murine myenteric plexus. (**A**) Photomicrographs of enteric GPR120 immunofluorescence (red) in wild-type and Thy1-αSyn mice maintained on different diets. Scale bar = 15 µm. (**B**) Quantification of GPR120 by immunoreactivity shows no difference between animals or among dietary groups (one-way ANOVA: *p* = 0.7613). (**C**) Photomicrographs of the myenteric plexus from Thy1-αSyn mice on control diet showing GPR120 (red) and TH^+^ neurons (green) reveal colocalization between the receptor GPR120 and TH^+^ neuronal body in the enteric nervous system, as indicated by the white arrow. Scale bar = 30 µm. Abbreviations: A.U., arbitrary unit; CTRL, control; DHA, docosahexaenoic acid; GPR120, G-protein coupled receptor 120; TH, tyrosine hydroxylase; WT, wild-type. (white dots) show an increase of varicosities per 100 μm of dendrites in transgenic mice fed with DHA-enriched diet compared to control diet.

**Figure 4 nutrients-13-04218-f004:**
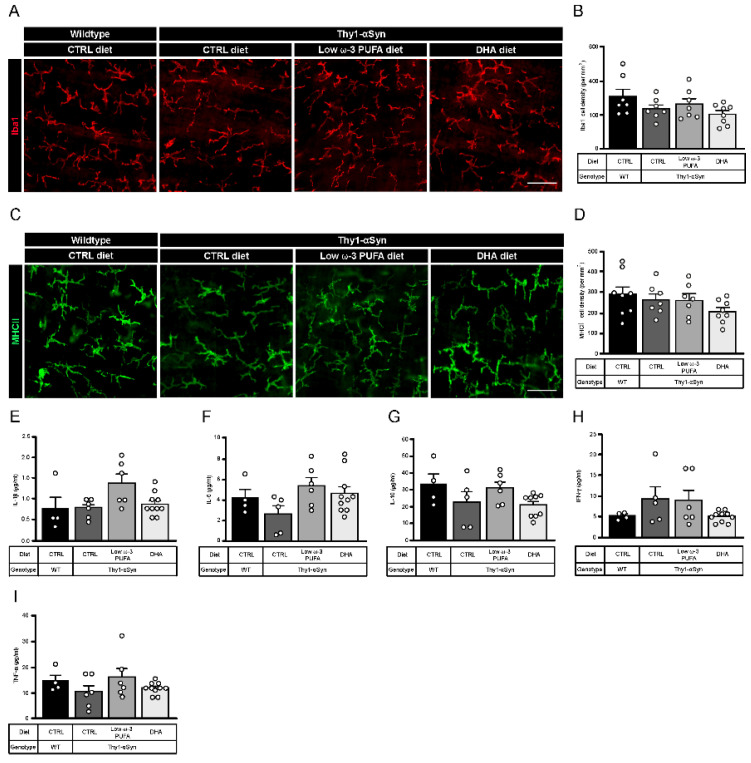
Thy1-αSyn mice do not show change in enteric macrophages and plasma cytokines composition. (**A**) Photomicrographs of Iba1^+^ macrophage immunofluorescence (red) in wild-type mice and the Thy1-αSyn model maintained on different diets. Scale bar = 30 μm. (**B**) Quantification of Iba1^+^ macrophages by density shows no significant difference between each group (one-way ANOVA: *p* = 0.1071). (**C**) Photomicrographs of MHCII^+^ macrophage immunofluorescence (green) in wild-type mice and the Thy1-αSyn model maintained on different diets. Scale bar = 30 μm. (**D**) Quantification of MHCII^+^ macrophages by density shows no significant difference between each group (one-way ANOVA: *p* = 0.2537). (**E**–**I**) Plasma cytokines analysis reveals no difference between mice strains and diets for IL-1β (**E**), IL-5 (**F**), IL-10 (**G**), IFN-γ (**H**) and TNF-α (**I**). Abbreviations: CTRL, control; DHA, docosahexaenoic acid; Iba1, ionized calcium-binding adapter molecule 1; IFN-γ, interferon gamma; IL, interleukin; MHCII, major histocompatibility complex class II; TNF-α, tumor necrosis factor alpha; WT, wild-type. (white dots) show an increase of varicosities per 100 μm of dendrites in transgenic mice fed with DHA-enriched diet compared to control diet.

**Figure 5 nutrients-13-04218-f005:**
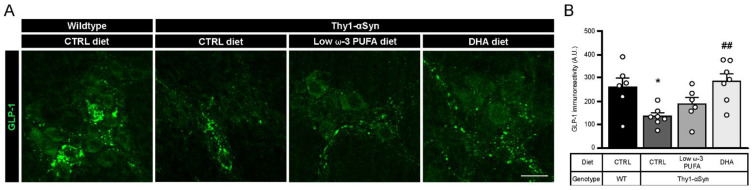
Increase in GLP-1 levels by a high DHA diet. (**A**) Immunofluorescence of GLP-1 in wild-type and Thy1-αSyn mice reveals the presence of the incretin in the myenteric plexus. Scale bar = 60 μm. (**B**) Quantification of the peptide revealed a lower expression in Thy1-αSyn mice compared with wild-type littermates. Transgenic mice fed with DHA-enriched diet show a higher level of GLP-1 than mice fed the control diet (one-way ANOVA: *p* = 0.0058). Tukey’s post-hoc tests: * *p* < 0.05 compared with wild-type mice; ^##^
*p* < 0.01 compared with Thy1-αSyn on control diet. Abbreviations: A.U., arbitrary unit; CTRL, control; DHA, docosahexaenoic acid; GLP-1, glucagon-like peptide-1; WT, wild-type. (white dots) show an increase of varicosities per 100 μm of dendrites in transgenic mice fed with DHA-enriched diet compared to control diet.

**Figure 6 nutrients-13-04218-f006:**
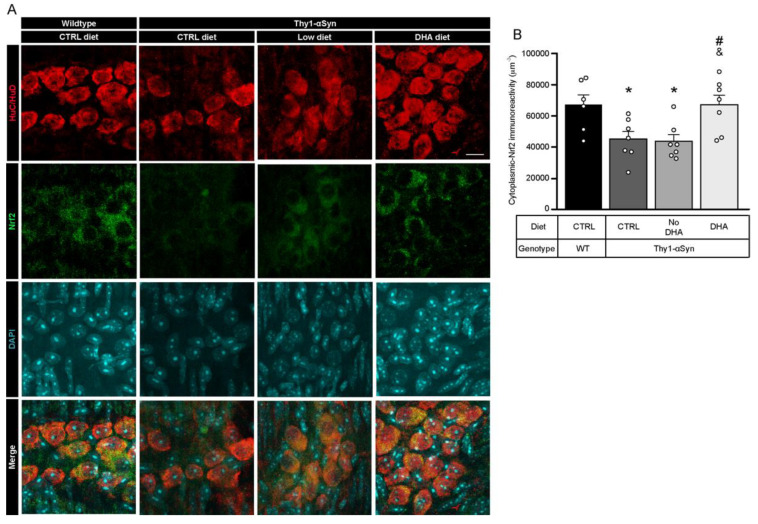
Augmentation of Nrf2 levels is associated with DHA-enriched diet. (**A**) Photomicrographs of HuC/HuD (red) and Nrf2 (green) immunofluorescence show colocalization of HuC/HuD and Nrf2 in the myenteric ganglions of Thy1-αSyn mice and their wild-type littermates. Maximum intensity projections are shown. Scale bar = 10 μm. (**B**) Immunoreactivity quantification of Nrf2 colocalizing with HuC/HuD (enteric neuron body) reveals lower levels of cytoplasmic-Nrf2 for the transgenic model on control and low ω-3 PUFA diets compared with wild-type animals. In contrast, mice maintained on the DHA-enriched diet show an increased Nrf2 level similar to wild-type mice (one-way ANOVA: *p* = 0.0040). Tukey’s post-hoc tests: * *p* < 0.05 compared with wild-type mice; ^#^
*p* < 0.05 compared with Thy1-αSyn on control diet; ^&^
*p* < 0.05 compared with Thy1-αSyn on the low ω-3 PUFA diet. Abbreviations: A.U., arbitrary unit; ChAT, choline acetyltransferase; CTRL, control; DHA, docosahexaenoic acid; Nrf2, nuclear (erythroid-2) related factor-2; TH, tyrosine hydroxylase; WT, wild-type. (white dots) show an increase of varicosities per 100 μm of dendrites in transgenic mice fed with DHA-enriched diet compared to control diet.

**Figure 7 nutrients-13-04218-f007:**
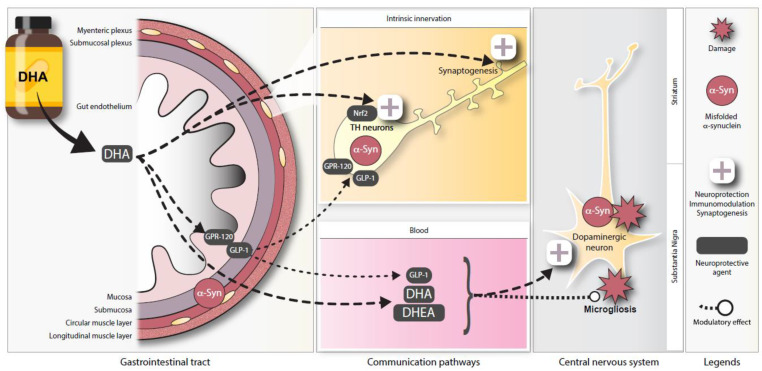
Schematic representation of potential neuroprotective pathways for DHA in the ENS. Illustration depicting a proposed model for DHA-mediated enteric neuroprotection in the Thy1-αSyn mouse model. Abbreviations: DHA, docosahexaenoic acid; DHEA, docosahexaenoyl ethanolamide; GLP-1, glucagon-like peptide-1; GPR120, G-protein coupled receptor 120; Nrf2, nuclear (erythroid-2) related factor-2; TH, tyrosine hydroxylase.

**Table 1 nutrients-13-04218-t001:** Plasma fatty acid composition in WT and Thy1-αSyn mice.

Fatty Acids (*p*mol/mL)		WT	Thy1-αSyn (Tg)			Mean Testing *p*-Values				
		CTRL	CTRL	Low n-3 PUFA	DHA	ANOVA	WT vs. Tg	CTRL vs. Low	CTRL vs. DHA	Low vs. DHA
ω-3 PUFA										
DHA	22:6n3	12.3 ± 0.6	16.7 ± 2.1	3.5 ± 0.8	162.2 ± 9.0	**** ^†^	>0.9999	**	**	****
n-3 DPA	22:5n3	2.1 ± 0.1	4.2 ± 0.5	15.3 ± 2.4	21.7 ± 1.5	**** ^†^	0.0877	0.1952	**	>0.9999
EPA	20:5n3	0.34 ± 0.02	0.49 ± 0.05	0.10 ± 0.02	4.71 ± 0.29	**** ^†^	0.6743	**	*	****
SDA	18:4n3	0.030 ± 0.006	0.121 ± 0.025	0.025 ± 0.009	0.184 ± 0.016	**** ^‡^	****	**	*	****
ω-6 PUFA										
AA	20:4n6	13.8 ± 0.8	15.5 ± 1.6	18.6 ± 1.2	8.8 ± 0.4	**** ^†^	>0.9999	0.6492	***	****
LA	18:2n6	12.7 ± 0.9	22.0 ± 2.4	23.9 ± 1.6	13.6 ± 0.6	**** ^‡^	****	0.8281	***	****

Abbreviations: AA, arachidonic acid; DHA, docosahexaenoic acid; DPA, docosapentaenoic acid; EPA, eicosapentaenoic acid; LA, linolenic acid; PUFA, polyunsaturated fatty acids; SDA, stearidonic acid; WT, wild-type. Values are represented as mean ± SEM (C57, *n* = 29; Tg-CTRL, *n* = 13; Tg-Low, *n* = 9; Tg-DHA, *n* = 18). Differences between genotype and between diet in transgenic mice were compared by ^†^ Kruskal–Wallis test with Dunn’s multiple comparisons or ^‡^ one-way ANOVA with Tukey post-hoc. Significant differences in means: * *p* < 0.05, ** *p* < 0.01, *** *p* < 0.001 and **** *p* < 0.0001.

**Table 2 nutrients-13-04218-t002:** Plasma endocannabinoid-like mediators in WT and Thy1-αSyn mice.

eCB-Like Mediators (*f*mol/μL)		WT	Thy1-αSyn (Tg)			Mean Testing *p*-Values				
		CTRL	CTRL	Low n-3 PUFA	DHA	ANOVA	WT vs. Tg	CTRL vs. Low	CTRL vs. DHA	Low vs. DHA
*N*-acyl-ethanolamine										
ω-3 PUFA derived										
DHEA	22:6n3	2.3 ± 0.2	2.0 ± 0.4	0.4 ± 0.2	5.8 ± 0.3	**** ^†^	ns	ns	****	****
DPEA (n-3)	22:5n3	0.2 ± 0.1	0.2 ± 0.1	2.2 ± 0.4	1.4 ± 0.3	**** ^†^	ns	**	*	ns
ω-6 PUFA derived										
AEA	20:4n6	3.5 ± 0.3	2.6 ± 0.3	2.3 ± 0.3	1.4 ± 0.1	**** ^‡^	ns	ns	*	ns
DPEA (n-6)	22:5n6	0.25 ± 0.07	0.6 ± 0.27	2.52 ± 0.55	2.50 ± 0.21	**** ^†^	ns	*	***	ns
LEA	18:2n6	7.2 ± 0.3	8.1 ± 0.6	9.7 ± 0.6	4.8 ± 0.2	**** ^†^	ns	ns	***	****
ω-9 MUFA derived										
OEA	18:1n9	19 ± 1	19 ± 1	15 ± 1	8 ± 0.2	**** ^†^	ns	ns	****	*
				SFA derived						
PEA	16:0	24 ± 1	20 ± 2	21 ± 1	22 ± 1	ns ^†^	ns	ns	ns	ns
SEA	18:0	41 ± 2	40 ± 4	35 ± 2	33 ± 2	ns ^†^	ns	ns	ns	ns
Monoacylglycerol										
ω-3 PUFA derived										
½-DHG	22:6n3	44 ± 5	74 ± 16	24 ± 5	369 ± 47	**** ^†^	ns	ns	**	****
½-EPG	20:5n3	1.2 ± 0.4	2.1 ± 0.9	n.d.	36 ± 4	**** ^†^	ns	ns	****	****
½-SDG	22:6n3	0.95 ± 0.12	0.60 ± 0.13	0.47 ± 0.09	0.7 ± 0.12	ns ^†^	ns	ns	ns	ns
ω-6 PUFA derived										
½-AG	20:4n6	0.31 ± 0.03	0.48 ± 0.09	0.93 ± 0.18	0.54 ± 0.07	**** ^†^	ns	ns	ns	ns
½-LG	18:2n6	170 ±60	620 ± 130	1020 ± 240	570 ± 60	**** ^†^	**	ns	ns	ns
ω-9 MUFA derived										
½-OG	18:1n9	1800 ± 100	3100 ± 500	3100 ± 600	2300 ± 300	* ^‡^	*	ns	ns	ns
SFA derived										
½-DPG	16:0	80 ± 11	158 ± 45	118 ± 26	162 ± 31	ns ^†^	ns	ns	ns	ns

Abbreviations: AEA, anandamide; AG, arachidonoyl glycerol; DHEA, docosahexaenoyl ethanolamide; DHG, docosahexaenoyl glycerol; DPEA, docosapentaenoyl ethanolamide; DPG, dipalmitoyl glycerol; EPG, eicosapentaenoyl glycerol; LEA, linoleoyl ethanolamide; LG, linoleoyl glycerol; ns, not significant; MUFA, monounsaturated fatty acid OEA, oleoyl ethanolamide; OG, oleoyl glycerol; PEA, palmitoyl ethanolamide; PUFA, polyunsaturated fatty acid; SEA, stearoyl ethanolamide; SDG, 1-stearoyl-2-docosahexaenoyl glycerol; SFA, saturated fatty acid; WT, wild-type. Values are represented as mean ± SEM (C57, *n* = 29; Tg-CTRL, *n* = 13; Tg-Low, *n* = 9; Tg-DHA, *n* = 18). Differences between genotype and between diet in transgenic mice were compared by ^†^ Kruskal-Wallis test with Dunn’s multiple comparisons or ^‡^ one-way ANOVA with Tukey post-hoc. Significant differences in means: * *p* < 0.05, ** *p* < 0.01, *** *p* < 0.001 and **** *p* < 0.0001.

## Data Availability

The datasets used and/or analyzed during the current study are available from the corresponding author on reasonable request.

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
