# Peer review of "Effect of Docosahexaenoic Acid (DHA) at the Enteric Level in a Synucleinopathy Mouse Model"

_nutrients, 2021, doi:10.3390/nu13124218_

Round 1

Reviewer 1 Report

The paper by Jérôme Lamontagne-Proulx describes DHA-enriched diet enhance neuropeotection of enteric DAergic neurons. Furthermore, the authors showed the mechanisms were related to GLP-1 pathway and Nrf2 pathway. The authors have some novel data but some of the methods need to be clarified.

  1. Please describe abbreviation TH+ (line 224)
  2. What is green stained with? (figure1)
  3. Please describe what kind of mouse is represented in figure 3C?
  4. Figure 3c represent just colocalization between GRp120 and TH+ neuronal. Did the colocalization change between genotype or diet group? The contribution is just one possibility as long as colocalization is changed. (line 293)
  5. Iba1+ immunoreactivity was also decreased in low ω-3 PUFA group. What is that explain by? Why did not figure 2S C suggest ω-3 PUFA effect on anti-inflammation? (line 316)
  6. What is HuC/HuD? (line356)
  7. Please make sure Nrf2 activation and alteration of oxidative stress marker. Figure 5 just represented expression. The data is not enough to determine oxidative stress was decreased by Nrf2. (line 363-364)
  8. Please describe name of ω-3 PUFA and ω-6 PUFA -derived eCB-like metabolites in table 2. (line 380-383)

Reviewer 2 Report

The manuscript entitled “Effect of Docosahexaenoic Acid (DHA) at the Enteric Level in a Synucleinopathy Mouse Model” (ID nutrients-1433058) by Lamontagne-Proulx et al. is a well written and interesting paper presenting results of evaluation of the effect of dietary supplementation with docosahexaenoic acid on the enteric nervous system. Authors concluded that dietary intake of DHA can protect  enteric DAergic neurons from synucleinopathy-induced DAergic neurodegeneration. In addition authors suggest that observed protective effects are associated with the levels of Nrf2 and w3-PUFAS containing eCB-like mediators. Since Parkinson’s disease involves aggregation of αSyn in the ENC, results presented in this article will be of interest to many not only in the field of utilization of w-3 PUFA in the supplementation, but also PD therapy.

The obtained results are extensively presented. However, there are some points that could be improved:

  • In materials and methods there is a statement that the dose of DHA was approximately 0.8 g/kg/daily per animal. How the DHA dose was established?
  • Some details in the methods section related to endocannabinoidome mediators are missing and should be provided:
    1. How the lipid metabolites were quantified? Were the calibration curves constructed?
    2. Were internal standards included in the method for quantification?
    3. The specific MRM mas transitions should be given to all determined lipid metabolites.

Round 2

Reviewer 1 Report

The authors have made substantial revisions in response to the original critique, and the manuscript is greatly improved as a result. However, there are still unclear points.

Point 8: Please describe name of ω-3 PUFA and ω-6 PUFA -derived eCB-like metabolites in table 2. (line 380-383)

Response 8: We do not understand this comment since all eCB-like metabolites are already described in the table 2 legend and materials and methods.

Response to response 8

I was not able to identify w-6 PUFA-derived metabolites in table 2. Please clarify what mediators are included in eCB-like metabolites or w-6 PUFA-derived metabolites. AEA is one of eCB-like metabolites, but it was decreased in DHA fed mice. what do you explain?
